# Clinical Practice of Targeted Capture Sequencing to Identify Actionable Alterations in Cholangiocarcinoma

**DOI:** 10.3390/cancers14205062

**Published:** 2022-10-16

**Authors:** Kai Ma, Youpeng Wang, Yuanzheng Zhang, Hongfa Sun, Xuhui Zhang, Chuandong Sun, Bingyuan Zhang, Ying Zhang, Haoyue Cheng, Ao Liu, Mengyao Wang, Bing Han

**Affiliations:** 1Department of Hepatobiliary and Pancreatic Surgery, The Affiliated Hospital of Qingdao University, Qingdao 266003, China; 2Collage of Medicine and Biological Information Engineering, Northeastern University, Shenyang 110169, China; 3School of Business Administration, Northeastern University, Shenyang 110169, China; 4Department of Pathology, Beijing Chaoyang Hospital, Capital Medical University, Beijing 100020, China; 5The Sixth Medical Center, Chinese PLA Medical School, Beijing 100853, China; 6Department of Research and Development, Shenzhen Byoryn Technology Co., Ltd., Shenzhen 518122, China

**Keywords:** cholangiocarcinoma, capture-based targeted sequencing, actionable genetic alterations, mutation landscape, biomarker

## Abstract

**Simple Summary:**

Genetic testing promises to provide guidance for early diagnosis and treatment of cholangiocarcinoma (CCA). Due to the different mutation landscapes across populations and the paucity of sequencing data of Chinese patients with CCA, the existing mutation landscape is insufficient to reflect the mutation characteristics of Chinese patients. We retrospectively analyzed 72 Chinese patients with CCA who had received genetic testing of targeted capture sequencing. A total of 2152 somatic mutations were detected in 56 (77.78%, 56/72) patients, of which, the frequently mutated driver genes were *TP53*, *KMT2D*, *KMT2C*, *BCOR*, *APC*, *BAP1*, *ARID1A*, *NF1*, *PIK3CA*, *KRAS*, and *LRP1B*. Most mutations were enriched in NRF2, TP53, and TGF-Beta oncogenic signaling pathways and cadherin repeat domain which were associated with intercellular adhesion. We identified 118 novel pathogenic or likely pathogenic somatic mutations and 77 actionable alterations, which provided potential targets for early diagnosis and treatment planning of CCA.

**Abstract:**

The early diagnosis and treatment of cholangiocarcinoma (CCA) remain a challenge worldwide. Genetic testing promises to solve these problems. Due to the different mutation landscapes across populations and the paucity of sequencing data of Chinese patients with CCA, the existing mutation landscape is insufficient to reflect the mutation characteristics of Chinese patients. Thus, we retrospectively analyzed 72 Chinese patients with CCA who had received genetic testing of targeted capture sequencing. A total of 2152 somatic mutations were detected in 56 (77.78%) patients, of which, the frequently mutated driver genes were *TP53* (27.78%), *KMT2D* (23.81%), *KMT2C* (20.63%), *BCOR* (18.06%), *APC* (15.28%), *BAP1* (13.89%), *ARID1A* (12.50%), *NF1* (12.50%), *PIK3CA* (12.50%), *KRAS* (11.11%), and *LRP1B* (11.11%). Most mutations were enriched in NRF2, TP53, and TGF-Beta oncogenic signaling pathways and cadherin repeat domains which were associated with intercellular adhesion. Based on cancer-related public databases and multiple protein function prediction algorithms, we identified 118 novel pathogenic or likely pathogenic somatic mutations and 77 actionable alterations. Molecular analysis of tumors from a precision oncology perspective can provide potential targets for early diagnosis and treatment of CCA and assist physicians in clinical decision making.

## 1. Introduction

Cholangiocarcinoma (CCA) is a malignant tumor arising from the epithelial cells of the bile ducts, which is the second most common liver cancer after hepatocellular carcinoma [1,2,3,4,5,6]. In the past 40 years, the overall incidence of CCA has gradually increased worldwide, and it is more common in Southeast Asia [7]. CCA is an invasive tumor with a poor prognosis [8]. More than half of the patients have locally advanced or metastatic disease at the time of presentation with a less than 30% five-year survival rate [9,10]. Early diagnosis and treatment of CCA remain a common challenge worldwide due to its “silent” clinical features and high heterogeneity, respectively [11].

With the continuous development and decreasing costs of sequencing technology, genetic testing is being increasingly applied toward the diagnosis and therapy of tumors clinically [12]. Many recent studies have shown that genetic testing for the individual patient with CCA has the potential to facilitate early diagnosis and guide patient-specific treatment [13,14]. Wardell et al. [15] carried out a large-scale genome sequencing analysis on 412 CCA patients from Italy and Japan and found 32 significant and common mutant genes, including *TP53*, *KRAS*, *SMAD4*, *NF1*, *ARID1A*, *PBRM1*, and *ATR*. Jusakul et al. [16] analyzed the comprehensive genomes, epigenomes, and transcriptomes of nearly 500 CCA patients from nine countries and identified novel driver genes (*RASA1*, *STK11*, *MAP2K4*, *SF3B1*) and structural variants (FGFR2 3’ UTR deletion). Wardell et al. [15] carried out a large-scale genome sequencing analysis on 412 CCA patients from Italy and Japan and found 32 significant and common mutant genes, including *TP53*, *KRAS*, *SMAD4*, *NF1*, *ARID1A*, *PBRM1*, and *ATR*. Jusakul et al. [16] analyzed the comprehensive genomes, epigenomes, and transcriptomes of nearly 500 CCA patients from nine countries and identified novel driver genes (*RASA1*, *STK11*, *MAP2K4*, *SF3B1*) and structural variants (*FGFR2* 3’ UTR deletion). Harmful germline mutations of cancer susceptibility genes such as *BRCA1*, *BRCA2*, *RAD51D*, *MLH1*, or *MSH2* were also detected in CCA patients. Fibroblast growth factor receptor (*FGFR*) fusions [17,18] and the repeated mutations of *IDH1* [19,20] were molecular features of the CCA subtype. Churi et al. [21] analyzed targeted sequencing-based mutation in 75 US patients with CCA and referred those with targetable mutations to appropriate clinical trials, further highlighting the important role of genetic testing in the diagnosis and treatment.

The European Society of Medical Oncology suggests that genetic testing should be routinely used in tumor samples of CCA to identify disease subgroups with different therapeutic implications and prognoses [12]. The common mutations of the same cancer varied among ethnic populations [22,23]. However, the aforementioned studies of CCA only recruited 18 Chinese patients, which means that the existing mutation landscape is insufficient to reflect the mutation characteristics of the Chinese patients. To explore and identify specific mutations of diagnostic and therapeutic significance in patients with CCA in China, this study was a retrospective review of 72 patients with CCA who had received genetic testing of targeted capture sequencing. We obtained CCA its mutation spectrum and frequently mutated genes, e.g., *BCOR*, *FAT3*, *APC*, and identified 118 novel pathogenic or likely pathogenic somatic mutations, which provided potential targets for early diagnosis and treatment planning of CCA.

## 2. Materials and Methods

### 2.1. Clinical Data and Samples Collection

We collected tumor tissue from the primary site and blood samples (as normal controls) from 72 Chinese Han patients with CCA from 2018 to 2021. Clinical diagnosis was confirmed by postoperative histopathological examinations. The collected tumor tissues were mainly tumor parenchyma, which was dominated by cholangiocarcinoma cells, and a small amount of tumor stroma. Clinical data and disease progression data were collected by the corresponding clinicians (Appendix A). The Ethics Committee of Affiliated Hospital of Qingdao University approved the research (approval no. QYFY WZLL 27314). All patients provided signed informed consent. We experimented according to the official guidelines issued by the National Health and Family Planning Commission of China.

### 2.2. DNA Extraction and Quality Evaluation

Tumor tissues were lysed by lysis buffer and incubated at 56 °C. After centrifuging and cooling, the supernatant is obtained, and after the transfer, an equal volume of phenol/chloroform/isoamyl alcohol (25:24:1) supernatant is added, and the primary treatment supernatant is obtained after repeated centrifugation. Isopropanol supernatant was added to the primary supernatant. The supernatant was removed after precipitation and the DNA particles were obtained after centrifugation and washing with 75% ethanol. The DNA particles were air-dried and dissolved by adding a buffer. DNA concentration was detected by a qubit fluorescence meter. The integrity and purity of samples were detected by agarose gel electrophoresis. DNA was extracted from peripheral blood leukocytes following similar steps as described above to filter germline mutations.

### 2.3. Library Construction and Sequencing

After DNA isolation, PCR amplifications were applied to establish the pre-capture library. The genomic DNA was first randomly fragmented using Covaris, followed by the screening of fragmented DNA with an average size of 200–400 bp using the Agincourt AMPure XP-Medium kit, and the screened fragments were subjected to end repair, 3′ adenylation, linker ligation, and targeted PCR amplification. Then, target enrichment and capture were performed with two custom sequence capture probes (Nimblegen, USA) that targeted 7708 exons of 508 cancer-related genes and 10,176 exons of 688 cancer-related genes, respectively. The targeted genes included proto-oncogenes and tumor-suppressor genes, genes with high-mutation frequency in solid tumors, tumor signaling pathway genes, drug-targeted genes, chemotherapy-related genes, and immunotherapy efficacy-related genes (For further details see Appendix A). During sample collection, the 508-gene panel was upgraded to a 688-gene panel. Thus, of these 72 patients, 63 were tested for 688 genes and nine were tested for 508 genes. A certain amount of PCR products were hybridized with BGI hybridization and a washing kit. The double-stranded PCR products were heat-denatured and looped through a splint oligonucleotide sequence to form single-stranded circular DNA (ssCirDNA) as the final library. On the BGISEQ-500 platform, combined probe-anchored synthesis (cPAS) was used for sequencing, and DNB was loaded into the patterned nanoarray to generate 100-bp pair-end reads.

### 2.4. Sequencing Data Analysis

After obtaining raw sequencing data, SOAPnuke [24] was used to remove adapters and filter low-quality reads. Clean reads were mapped to the human genome (hg38) by using bwa-mem2 (https://github.com/bwa-mem2/bwa-mem2/, accessed on 16 October 2020) [25]. GATK (v 4.1.9.0) [26] was used to remove duplicates (MarkDuplicates module), call somatic variants (Mutect2 module), and filter variants (FilterMutectCalls module). Sequencing data from paired blood samples were used to assess for germline mutations in order to better identify true somatic mutations. CalculateContamination and LearnReadOrientationModel of GATK were used to assess cross-sample contamination and read orientation bias to filter variants, respectively. ANNOVAR (https://annovar.openbioinformatics.org/en/latest/, accessed on 2 February 2021) [27] was performed for variants annotation, including interpreting clinical significance and predicting the functional impact of sequence variant/linkage alterations. Next, somatic variants were filtered according to the following criteria: (i) variants beside exonic or splicing region were filtered out; (ii) variants with allele frequency (AF) < 0.1 were filtered out; (iii) variants with population frequencies > 1% were excluded from further analysis according to the Exome Aggregation Consortium dataset (ExAC, https://gnomad.broadinstitute.org/, accessed on 8 July 2021), 1000 Genomes Project (http://www.1000genomes.org/, accessed on 30 September 2021) [28], ESP6500SI-V2 and avsnp150 database. Moreover, OncoKB (http://oncokb.org/, accessed on 29 March 2022) [29] was employed to identify actionable mutations. IntOGen (https://www.IntOGen.org/, accessed on 1 February 2021) [30] offered CCA driver genes in Hartiwig (https://www.hartwigmedicalfoundation.nl/en/data/database/, accessed on 1 February 2021), International Cancer Genome Consortium (ICGC), the Cancer Genome Atlas (TCGA), and Pan-Cancer Analysis of Whole Genomes (PCAWG) cohorts.

### 2.5. Mutation Statistics and Visualization

R package maftools (version 2.8.05) [31] were used to summarize and visualize mutation information, including mutation signature, mutation distribution, and oncogenic signaling pathways enriched. We organized the annotated mutation information into CSV files following the Oviz-Bio platform landscape analysis (https://bio.oviz.org/demo-project/analyses/landscape/, accessed on 2 July 2020) [32], characterizing the mutation type, mutation gene, mutation frequency, and clinical information of the patient. Fisher’s Exact test or χ² tests were used to compare the mutation frequencies between our study and the IntOGen cohort and analyze the associations between driver genes and clinical characteristics were analyzed. *p*-value less than 0.05 was considered meaningful.

## 3. Results

### 3.1. Clinical Characteristics of the Patients with CCA

The study involved 72 patients with CCA (25 females and 43 males). The median age of the patients was 61.41 (range 28–83). Half of the patients were hilar CCA, 26.39% were distal extrahepatic CCA, and 16.67% were intrahepatic CCA. At the same time, most patients (63.89%, 46/72) were in the T2 phase. More than half of the patients had a tumor diameter between 0–3 cm, while 33.34% (24/72) of patients had a tumor diameter > 3 cm. Nerve infiltration is common in CCA and is considered to be a crucial step in tumor metastasis. Overall, 37 patients included 31 cases with nerve infiltration and eight cases with lymph node metastasis. Among them, two cases had both nerve infiltration and lymph node metastasis. Additionally, 20 patients (27.78%) relapsed.

As for clinical indices, serum ASL/ALT ratios were less than 0.8 and more than 1.5 in six and 42 patients, respectively. ASL/ALT ratio can help to determine whether the patient has some liver or bile duct disease. CA19-9 and CA50, as the first choice markers of CCA, are the important basis for judging the clinical condition of patients. The CA19-9 and CA50 levels were above the upper limit of normal in 51 and 43 patients, respectively, while AFP, CEA, and CA125 were within normal values in most of the patients (Table 1 and Appendix A).

### 3.2. The spectrum of Somatic Mutations in Shared Genes

Of these 72 patients, 63 were tested for 688 genes and nine were tested for 508 genes. There were a total of 850 targeted genes captured by 688- and 508-gene panels, including 345 shared genes, 343 genes specific in the 688-gene panel, and 163 genes specific in the 508-gene panel. In total, we detected 2152 somatic mutations in 56 (77.78%, 56/72) patients (Appendix A), including 399 SNVs, 23 DNVs, 1128 insertions, and 602 deletions (Figure 1a,b). C > T and C > A alteration were the major mutant forms (Figure 1c). In addition, there were 60 synonymous mutations, which we filtered in the following analysis.

We first observed mutations in 345 genes shared by 688- and 508-gene panels (Figure 2a and Appendix A). Among them, 245 shared genes were mutated in 56 patients, ~70% of which were mutated in only 1–2 patients (Figure 1c). We got 45 mutated CCA driver genes identified by TCGA, ICGC, PCAWG, and Hartiwig cohorts and found 35 of them mutated in 55 patients in this study. Of these, the frequently mutated driver genes were *TP53* (27.78%, 22/72), *BCOR* (18.06%, 13/72), *APC* (15.28%, 11/72), *BAP1* (13.89%, 10/72), *ARID1A* (12.50%, 9/72), *NF1* (12.50%, 9/72), *PIK3CA* (12.50%, 9/72), and *KRAS* (11.11%, 8/72). For *TP53*, *ARID1A*, and *KRAS*, the mutation frequencies were not different from the above four cohorts (Fisher’s exact test; adjust-*p* > 0.05). However, the mutation frequencies of other driver genes varied between our study and the above four cohorts, such as *BCOR* (18.06% vs. 0.77%; adjust-*p* < 0.001), *APC* (15.28% vs. 1.79%; adjust-*p* < 0.001), *BAP1* (13.89% vs. 4.09%; adjust-*p* = 0.008), *NF1* (12.50% vs. 3.58%; adjust-*p* = 0.015), and *NCOR1* (8.33% vs. 0.51%; adjust-*p* < 0.001; Appendix A).

In addition to these known driver genes, *FAT3* (26.39%, 19/72), *SPEN* (19.44%, 14/72), *TET2* (15.28%, 11/72), *MECOM* (13.89%, 10/72), *MYC* (13.89%, 10/72), *EPPK1* (13.89%, 10/72), *ZNF217* (12.50%, 9/72), *PIK3CG* (12.50%, 9/72), *ARID1B* (12.50%, 9/72), and *CIC* (12.50%, 9/72) were also mutated frequently in our study (Figure 2a and Appendix A).

### 3.3. The Spectrum of Somatic Mutations in Genes Specific to 688- and 508-Gene Panels

Of 343 genes only detected by a 688-gene panel in 63 patients, 196 of them were mutated in 49 patients. As driver genes, *KMT2D* (23.81% vs. 0.51%; adjust-*p* < 0.001), *TRRAP* (9.52% vs. 0.77%; adjust-*p* < 0.001), *LRP1B* (12.70% vs 1.53%; adjust-*p* < 0.001), and *KMT2C* (20.63% vs. 2.81%; adjust-*p* < 0.001) were mutated more frequently than the above four cohorts (Appendix A). Moreover, *MUC16* (46.03%, 29/63), *FAT4* (44.44%, 28/63), *ZFHX4* (33.33%, 21/63), *APOB* (31.74%, 20/63), *FAT1* (31.74%, 20/63), *RYR2* (23.81%, 15/63), *KMT2A* (23.81%, 15/63), and *ZFHX3* (20.63%, 13/63) were mutated frequently in our study (Figure 2b and Appendix A).

For 163 genes specific in the 508-gene panel in nine patients, five mutations in *HIF1A*, *HDAC3*, *MSR1*, and *TAF1* were detected in four patients (Appendix A). Due to the small sample number, we did not detect frequently mutated genes, which were only targeted by the 508-gene panel.

### 3.4. Clinical Implications of Mutations

We annotated the clinical significance of these mutations based on the CLINVAR, dbSNP, COSMIC, and OncoKB databases and used 21 algorithms to predict the pathogenic mutation. There were 58, 99, and 71 variants registered in CLINVAR, dbSNP, and COSMIC databases, respectively (Figure 3a). These variants were considered to be functionally important mutations. Of the 457 novel variants which were not registered in dbSNP, ClinVar, and COSMIC70, 118 variants were predicted to be deleterious at least by five algorithms (Appendix A). We detected the most novo pathogenic mutations in *KMT2D* (Appendix A). Moreover, there were also 444, 19, and two variants reported as a loss of function, gain of function, and switch of function mutations, respectively. Seventy-seven variants in 40 patients were reported as the drug targets, such as *KRAS* G12D (n = 4, MEK inhibitors), *CTNNA1* exon 11 deletions (Hedgehog inhibitors), and *BRCA2* exon 11 deletions (PARP inhibitors) (Appendix A).

### 3.5. Effect of Mutations on Domains and Oncogenic Signaling Pathways

Next, we performed domain enrichment analysis and found that cadherin repeat (CA), cadherin-type repeats region, and P53 domains were affected more frequently (Figure 3b). Concretely, all variants in *TP53* were concentrated in the P53 structural domain, including 13 missense mutations, two in-frame insertions, and one each of frameshift insertion and frameshift deletion. Most patients carried the p.R209Q/W and p.R234H/C mutations (Figure 3c). In contrast, almost all mutations except for p.C1465X were located outside the Ankyrin domain of *BCOR* (Figure 3d). The situation was similar in *MUC16* and *KMT2D* (Appendix A).

Moreover, we observed that the oncogenic signaling pathways with the most mutated genes were RTK-RAS (38 mutated genes). Among the mutated genes, five were tumor suppressors and 29 were oncogenes (Figure 3e,f). The most affected pathways were NRF2, TP53, and TGF-Beta pathways, in which more than half of the genes mutated. In the TP53 pathway, TP53, ATM, CHEK2, and MDM4 were mutated (Appendix A).

### 3.6. Associations between Driver Genes and Clinical Characteristics

We also analyzed the associations between driver genes and clinical characteristics Smoking appeared to be associated with *ARID1A* mutations (*p* = 0.02). *TP53* mutations were not associated with age and sex in patients with CCA. We did not find any driver gene associated with nerve infiltration and lymph node metastasis (Appendix A).

## 4. Discussion

This retrospective study sheds light on the hotspot mutational profile of 72 patients with CCA in China. We found that 77.78% of patients harbored somatic mutations and 55.56% of patients harbored actionable alterations, which means targeted panel sequencing could assist physicians to develop emerging therapeutic strategies. Moreover, we identified 118 novel pathogenic or likely pathogenic somatic mutations, which could be potential tumor markers to improve the accuracy of early diagnosis.

In our study, the mutation frequencies of *TP53*, *ARID1A*, and *KRAS* were similar to those in TCGA, ICGC, PCAWG, and Hartiwig cohorts. An interesting finding is that some driver genes, such as *BCOR*, *APC*, *BAP1*, *NF1*, and so on, had higher mutation frequencies in Chinese patients. This might be due to the different genetic backgrounds [22,23]. In addition to these known driver genes, mutated genes with high frequency, such as *MUC16* (46.03%), *FAT4* (44.44%), *ZFHX4* (33.33%), *APOB* (31.74%), *FAT1* (31.74%), *FAT3* (26.39%), *RYR2* (23.81%), *KMT2A* (23.81%), and *ZFHX3* (20.63%) deserved more attention for their potential as biomarkers of early diagnosis. Moreover, we observed that C > T and C > A transversions were the more common substitution in the samples, which were confirmed by previous studies [33]. Nepal et al. concluded that C > A translocation is associated with oxidative stress, which was considered a major pathogenic factor in the progression of cancer [34].

About 65% of patients in this study had alterations in the RTK-RAS pathway. The activation of the RTK-RAS signaling pathway is a common event across all subtypes of CCA [35]. Cell cycle pathways, PI3K pathways, and Hippo pathways were affected frequently in our study. Mutations involved in these oncogenic signaling pathways can alter the intercellular adhesion of cadherins, making it closely associated with tumor metastasis, angiogenesis, and invasiveness [36]. Remarkably, most mutations were enriched in the CA domain. Cadherins are glycoproteins involved in Ca^2+^-mediated cell-cell adhesion through their extracellular structural domains. Previous studies have found that cadherins had important functions in cell-cell adhesion, tissue patterning, and carcinogenesis. Any dysfunction or instability of the cadherin-catenin complex may lead to tumor progression. Consistent with these findings, CCA is likely to develop invasion and metastases. On the other hand, we did not find any driver gene associated with nerve infiltration and lymph node metastasis. We think that the inadequate sample size of the cohort limited the statistical power. Therefore, the results of the associations between driver genes and clinical characteristics need to be interpreted with caution. Otherwise, targeted sequencing does not cover the whole genome. If a structural variation breakpoint lies in an untargeted region, it cannot be detected. Thus, we cannot detect and analyze the structural variation of patients with CCA.

## 5. Conclusions

In summary, our results presented a clear genomic landscape of the mutation frequencies of oncogenic drivers from 72 patients with CCA. Molecular analysis of tumors from a precision oncology perspective can provide potential targets for early diagnosis and treatment of CCA and assist physicians in clinical decision making. We hope that our study findings can promote the application of routine genetic testing in clinical practice. 

## Figures and Tables

**Figure 1 cancers-14-05062-f001:**
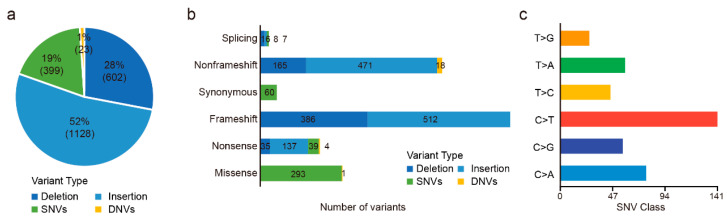
Outcomes of variants calling in 72 CCA patients. (**a**) Percentage of SNVs, DNVs, and indels. (**b**) Numbers of missense, nonsense, synonymous, frameshift, nonframeshift, and splicing variants. (**c**) Numbers of each SNV class.

**Figure 2 cancers-14-05062-f002:**
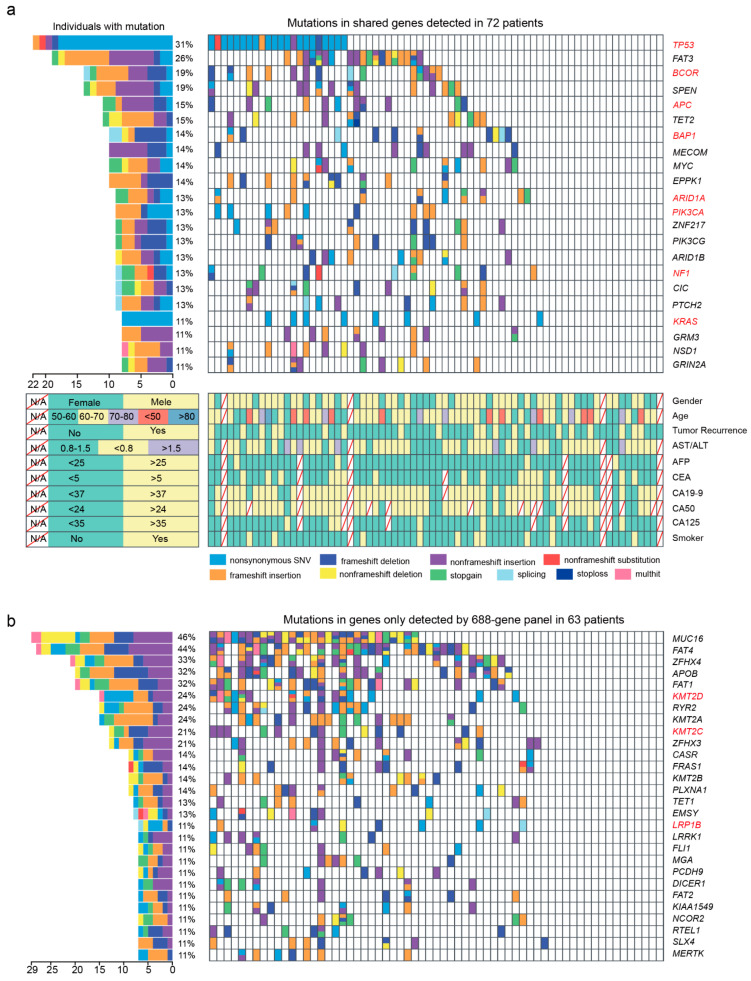
The landscape of frequently mutated genes of CCA. (**a**) Significantly mutated genes both detected by 508- and 688-gene panels in 72 patients. (**Left**), the percentages of patients with mutations. (**Right**), targeted genes are ranked based on the mutation frequency. Red letters indicate driver genes. Bottom, the clinical information of each patient. Diagonal line indicate information that was not available. Different colors correspond to different types of mutations. Variants annotated as Multhit are those genes that are mutated more than once in the same sample. (**b**) Mutation landscape of gene only detected by the 688-gene panel in 63 patients. Genes with a frequency greater than 10% are shown. CC: cholangiocarcinoma; ECC: extrahepatic cholangiocarcinoma; HCC: hilar cholangio-carcinoma; ICC: intrahepatic cholangiocarcinoma.

**Figure 3 cancers-14-05062-f003:**
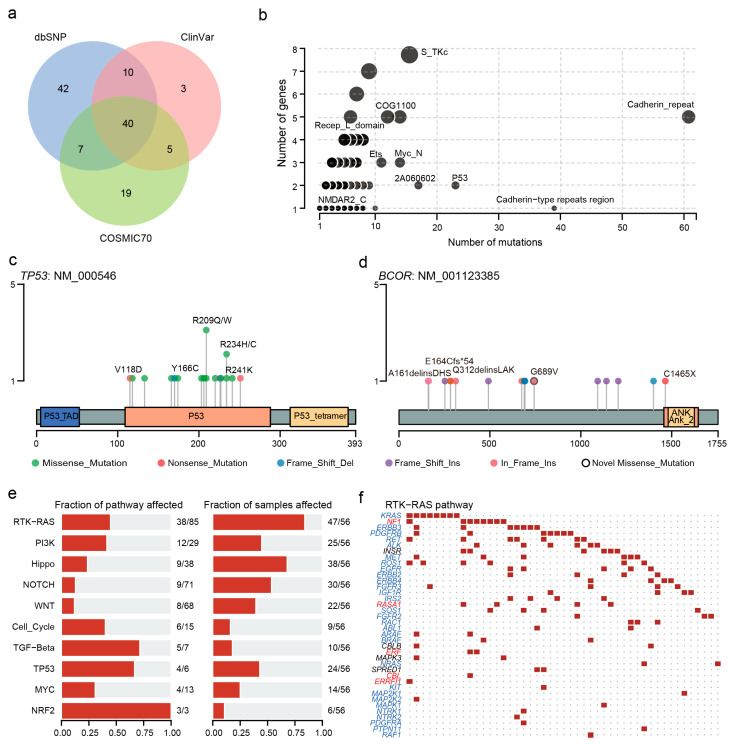
Clinical implications of mutations and domain and pathway enrichment analysis. (**a**) Venn plot of database-registered variants. (**b**) Frequently mutated pfam protein domains in CCA. The size of the bubble is in proportion to the number of genes containing prominent display domains. (**c**) Location of TP53 mutations schematic. (**d**) Location of the BCOR mutations schematic. (**e**) Enrichment of known oncogenic signaling pathways. (**f**) Mutated genes in the RTK-RAS pathway. Tumor suppressor genes are in red, and oncogenes are in blue font. The red square represents a patient with one or more mutations in this gene and the black dot represents no mutation detected in the patient.

**Table 1 cancers-14-05062-t001:** Clinical characteristics of 72 patients with CCA.

Clinical Characteristics (*n* = 72)	
Age years	
Median	61.41
Range	28–83
Sex	
Male	43 (59.72%)
Female Unknown	25 (34.72%) 4 (5.56%)
With Diabetes	
Yes	16 (22.22%)
No	52 (72.22%)
Unknown	4 (5.56%)
Smoker	
Yes No Unknown	10 (13.89%) 58 (80.56%) 4 (5.56%)
Drinker	
Yes No Unknown	14 (19.44%) 54 (75.00%) 4 (5.56%)
BMI	
18–24 24–27.5 >27.5	31 (43.06%) 23 (31.94%) 14 (19.44%)
Unknown	4 (5.56%)
Bismuth-Corlette Classification	
1	11 (15.28%)
2 3 a 3 b 4 Not involved Unknown	4 (5.56%) 6 (8.33%) 8 (11.11%) 7 (9.72%) 31 (43.06%) 5 (6.94%)
TNM Stage	
T1	1 (1.39%)
T2	46 (63.89%)
T3 T4 Unknown	8 (11.11%) 6 (8.33%) 11 (15.28%)
Tumor Diameter	
0–3 cm 3–5 cm 5–7 cm 7–10 cm Unknown	37 (51.39%) 17 (23.61%) 4 (5.56%) 3 (4.17%) 11 (15.28%)
Nerve Infiltration	
Yes No Unknown	31 (43.06%) 29 (40.28%) 12 (16.67%)
Lymph Node Metastasis	
Yes No Unknown	8 (11.11%) 52 (72.22%) 12(16.67%)
Postoperative Treatment	
Chemotherapy with Tegafur Interventional therapy immunotherapy No	49(68.06%) 1(1.39%) 2(2.78%) 20(27.78%)
Tumor Recurrence	
Yes No	20 (27.78%) 52 (72.22%)

## Data Availability

The datasets generated during and/or analyzed during the current study are available from the corresponding author on reasonable request.

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
