# Peer review of "Clinical Practice of Targeted Capture Sequencing to Identify Actionable Alterations in Cholangiocarcinoma"

_cancers, 2022, doi:10.3390/cancers14205062_

Round 1

Reviewer 1 Report

I thank the authors for the opportunity to review their paper detailing the genetic landscape of cholangiocarcinoma.

Overall, the research presented is of interest to the HPB cancer community. it is somewhat limited by its small cohort size and very selective gene panel, but it provides an interesting view of the mutational landscape of CCA in a specific populationcohort. 

Specific comments:

Materials and Method -

1. Clinical data and sample collection: The authors mention that the final diagnosis was based on cytopathology. Does that mean that formal histology was not assessed or taken into account when sanalyzing these cases?

2.  Clinical data and sample collection: "blood samples (as normal controls)" were collected. If these were used to assess for germline mutations in order to better identify true somatic mutations, please state so.

3. Library Construction and Sequencing: What was the basis for the selection of the library targets? Were these targets used elsewhere in other published studies? Were any targets used as controls/ housekeeping genes? Please provide the referred Table S2 - I was not able to access it.

 Results:

Table 1 - It seems from your use of the Bismuth classification that all of the cases were perihilar. is that correct? if not, how many were distal/extrahepatic?

I would recommend rearranging the table and moving co-morbid conditions (diabetes, smoking, alcohol abuse) to the beginning of the table after age and sex. Please also include BMI.

Discussion:

The authors state that "We found that 77.78% of patients harbored actionable genetic alterations and 55.56% of patients harbored drug-targeted mutations". However, They have not shown these 77.78% of mutations to be actionable (i.e, of clinical relevance for specifically tailored therapy). Similarly, the term "drug-targeted mutations" is probably misleading as well.  

Was there a difference in the mutational landscape based on disease stage?

Please provide better quality images for Figures S2, S3.

Reviewer 2 Report

The authors use BGI-seq (Beijing Genomics Institute platform) to perform targeted amplicon sequencing of 508 or 688 genes. 78% of patients had a mutation detectable on this platform. Out of 45 driver genes identified in other projects (e.g., TCGA), 35 were mutated in this study. A few genes, notably KMT2C and KMT2D, were mutated much more in this cohort than in published cohorts. These are usually mutated in lymphomas and it is difficult to believe a main oncogenic role in cholangiocarcinoma has been overlooked. I am skeptical that there are major differences in the genomic landscape between Chinese and non-Chinese populations. Using a different sequencing platform from Western studies may be part of the explanation, or there could be an exposure in the study population such as smoked meat/preserved food, although this is somewhat of a stereotype.

The study is not highly novel, but appears appropriately performed. The authors have leveraged a variety of tools to make inferences, representing a strength. 

MAJOR POINTS

The premise is that data on CCA mutations is needed in Chinese patients, but that is a very heterogeneous population. If we accept the premise that tumor mutations will be different in different ethnic groups, then more should be stated about the population. Are they Han Chinese, minority peoples, etc.?

There is a lack of data on the tumor samples. We know only that "we collected tumor tissue" (line 85). Was a certain neoplastic cellularity required? Were the tumors from primary sites (as I suspect)?

The Supplemental Tables were not provided for peer review (either in the main file or supplement) and should be provided to ensure they are present in the final published version.

Authors found 35 previously reported driver genes (out of 45) were mutated in their patients. Were the other 10 part of their capture region and did they have good coverage? Can we conclude that those drivers are NOT mutated in Chinese patients? Rare drivers from foreign populations might just not be represented in a 72 patient cohort. The authors can use Fisher's test for these genes as they do for the others. They can consider whether it is necessary to adjust the significance (alpha) level given that they are performing multiple comparisons.

The authors speculate at l. 266 that the observation that some driver genes were mutated here but not in other cohorts could be due to different genetic backgrounds or to a "not large enough cohort". Since the other cohorts are larger, this appears unlikely. 

MINOR POINTS

In 2.3 please clarify the library construction method and nature of the target region. It is not clear to me whether this was an amplicon method or something else, and whether the entire gene was targeted or only hotspots. 

At line 113, "100 base readings at the paired end" is non-idiomatic and should be rephrased.

Clarify why some patients were tested for different gene panels (line 166).

Figures should be submitted at higher resolution. They are blurry.

At line 192, "diagonally indicated the information" --> "Diagonal line indicates information".

Forgive me, but Section 3.3 is difficult to read and the point being made is not clear. I think the intention is to address the difference between the two panels. If this is important, please clarify. 

At line 229, "PAPR" --> "PARP".

Figure 3e, please clarify meaning of "fraction of pathway affected". Does this mean the fraction of genes in the pathway that are mutated in at least one sample? 

The method doesn't detect structural variants and the authors could comment on whether this is a limitation. 

Round 2

Reviewer 1 Report

The authors have addressed my comments in a satisfactory fashion.